# Developing Porous Ortho- and Pyrophosphate-Containing Glass Microspheres; Structural and Cytocompatibility Characterisation

**DOI:** 10.3390/bioengineering9110611

**Published:** 2022-10-25

**Authors:** Ben Milborne, Lauren Murrell, Ian Cardillo-Zallo, Jeremy Titman, Louise Briggs, Colin Scotchford, Alexander Thompson, Robert Layfield, Ifty Ahmed

**Affiliations:** 1Advanced Materials Research Group, Faculty of Engineering, University of Nottingham, Nottingham NG7 2RD, UK; 2School of Chemistry, University of Nottingham, Nottingham NG7 2RD, UK; 3Biodiscovery Institute, Division of Cancer and Stem Cells, University of Nottingham, Nottingham NG7 2RD, UK; 4School of Life Sciences, Faculty of Medicine and Health Sciences, University of Nottingham, Nottingham NG7 2UH, UK

**Keywords:** phosphate-based glasses, orthophosphates, pyrophosphates, bone repair, orthobiologic

## Abstract

Phosphate-based glasses (PBGs) are promising materials for bone repair and regeneration as they can be formulated to be compositionally similar to the inorganic components of bone. Alterations to the PBG formulation can be used to tailor their degradation rates and subsequent release of biotherapeutic ions to induce cellular responses, such as osteogenesis. In this work, novel invert-PBGs in the series xP_2_O_5_·(56 − *x*)CaO·24MgO·20Na_2_O (mol%), where *x* is 40, 35, 32.5 and 30 were formulated to contain pyro (Q^1^) and orthophosphate (Q^0^) species. These PBGs were processed into highly porous microspheres (PMS) via flame spheroidisation, with ~68% to 75% porosity levels. Compositional and structural analysis using EDX and ^31^P-MAS NMR revealed that significant depolymerisation occurred with reducing phosphate content which increased further when PBGs were processed into PMS. A decrease from 50% to 0% in Q^2^ species and an increase from 6% to 35% in Q^0^ species was observed for the PMS when the phosphate content decreased from 40 to 30 mol%. Ion release studies also revealed up to a four-fold decrease in cations and an eight-fold decrease in phosphate anions released with decreasing phosphate content. In vitro bioactivity studies revealed that the orthophosphate-rich PMS had favourable bioactivity responses after 28 days of immersion in simulated body fluid (SBF). Indirect and direct cell culture studies confirmed that the PMS were cytocompatible and supported cell growth and proliferation over 7 days of culture. The P30 PMS with ~65% pyro and ~35% ortho phosphate content revealed the most favourable properties and is suggested to be highly suitable for bone repair and regeneration, especially for orthobiologic applications owing to their highly porous morphology.

## 1. Introduction

The biocompatibility of phosphate-based glasses (PBGs) and the ability to tailor their formulation to be compositionally similar to the inorganic component of bone and control their degradation profile make them encouraging materials for hard tissue engineering applications [1]. The primary network former of pure phosphate glass is P_2_O_5_ but due to its hygroscopic nature, modifying oxides are added to improve the stability and properties of PBGs [2]. The phosphate tetrahedral groups are the structural unit of PBGs, with the P-O-P bonds that form between the adjacent tetrahedral known as bridging oxygens (BOs) [3]. The structure of PBGs is related to the number of BO’s and Q^n^ terminology is used to describe the species present within PBGs, where *n* signifies the number of BO’s per phosphate tetrahedron [4]. A range of phosphate glass anionic Q^n^ species can be produced from Q^3^, Q^2^, Q^1^ and Q^0^ species, which are referred to as ultra-, meta-, pyro- and orthophosphates, respectively.

PBGs generally degrade faster than silicate-based glasses and altering the glass composition can vary the rates of dissolution by several orders of magnitude [5]. The network connectivity and the arrangement of modifier cations within the glass structure can dictate the dissolution rates of PBGs. The addition of alkali (Me_2_O) and alkaline earth (MeO) oxides reduces the network connectivity of the PBGs by decreasing the number of BO’s between the phosphate tetrahedrons. The modifier oxides reside in the spaces between the phosphate tetrahedrons and the ionic strength of the modifiers determines the strength of the bond between them and the network former. The strength of these interactions can have a profound effect on the physical properties of the glass and its dissolution rate [6]. The addition of Me_2_O and MeO can be used to introduce a controlled amount of therapeutic ions into the phosphate glass composition during the standard manufacturing process which is subsequently released upon the glass degradation [7]. Careful formulation development can therefore be used to tailor the specific ions as well as the rate in which they are released from PBGs.

The relative ease at which various therapeutic ions can be doped into PBGs has led to numerous novel formulations being developed and investigated. Single inorganic ions, such as calcium (Ca), phosphorous (P) and magnesium (Mg) are vital for bone metabolism and for growth and mineralisation of bone tissue [8]. Calcium ions are known to promote osteoblast proliferation and differentiation and activate Ca-sensing receptors on osteoblasts to increase the expression of growth factors, such as IGF-I and IGF-II [9]. Magnesium is essential for the formation of the alkaline phosphatase (ALP) [10] enzyme, which has a vital regulatory role in bone mineralisation [11]. PBGs containing these ions will release them during their dissolution in vivo over time. These modifier cations, which are ionically linked to NBO’s, can be released from the glass structure in a physiological environment.

Me_2_O and MeO modifying oxide addition subsequently causes a reduction in the structural connectivity and can be used to tailor the phosphate species present with PBGs. Invert phosphate glasses, that typically contain less than 40 mol% P_2_O_5_, are primarily composed of pyro- and orthophosphate species and have been shown to have more controlled degradation rates than ultraphosphate PBGs [12]. Pyro- and orthophosphates are present within the body and are integral to the natural cycle of bone resorption and formation [13]. Although inorganic pyrophosphate (PPi) can inhibit mineralisation by preventing the attachment and crystal growth of hydroxyapatite (HA), it has also been demonstrated to promote ALP activity and gene expression [14]. ALP, which is secreted by active osteoblasts, can hydrolyse PPi into orthophosphates causing a loss of PPi inhibition on HA formation whilst simultaneously resulting in the local saturation of orthophosphates that induce mineralisation [15]. As such, the development of PBGs that contain these anionic phosphate species could hold significant potential for bone repair due to their intrinsic role in bone mineralisation.

PBG can also be manufactured into a variety of geometries depending on the desired application. Microspheres have been extensively investigated for biomedical applications due to their beneficial functional characteristics. Spheres exhibit greater uniformity in size and shape in comparison to irregular-shaped particles and have a greater surface area and enhanced flow properties. PBG microspheres have a more controlled and predictable rate of degradation and subsequent release of incorporated therapeutic ions as opposed to irregular-shaped particles [16]. In addition to this, the enhanced flow of microspheres provides improved delivery properties and the ability for them to be administered via minimally invasive injection procedures [17].

Microspheres can be manufactured to be solid, porous and hollow depending on their intended purpose. Porosity further increases surface area, which allows for greater cell attachment, encapsulation of bioactive compounds and the flow of nutrients and gases [18]. Various manufacturing processes, such as sol-gel and polymer foam replication, have been used to manufacture porous glass/glass-ceramic microspheres [19]. Many of these techniques are time-consuming and labour intensive due to their multi-step nature and the need to remove residual contaminants. Recent studies have shown that a novel single-stage flame spheroidisation method can be used to produce highly porous calcium phosphate glass microspheres [20].

This study reports on the manufacture of four quaternary PBGs in the series xP_2_O_5_·(56 − *x*)CaO·24MgO·20Na_2_O (mol%), where *x* is 40, 35, 32.5 or 30, and their production into solid (dense) and highly porous microspheres. The characterisation studies focused on their thermo-physical properties, ion release profiles and In vitro bioactivity via SBF studies. Further, porous microspheres from these PBG formulations were tested for their In vitro cytocompatibility, using human osteoblast-like cells (MG63s) as a clinically relevant cell type. Indirect and direct cell culture systems were tested to evaluate the materials’ ability to support cell growth and their osteogenic potential.

## 2. Materials and Methods

### 2.1. Phosphate Based Glass Fabrication

The phosphate glasses were prepared using the following precursors: sodium dihydrogen phosphate (NaH_2_PO_4_), calcium hydrogen phosphate (CaHPO_4_), calcium carbonate (CaCO_3_) and magnesium hydrogen phosphate trihydrate (MgHPO_4_·3H_2_O) (Sigma Aldrich, Gillingham, UK) (see Table 1 for formulations). The precursors were weighed according to the composition and mixed thoroughly before being heated in a 5% Au/Pt crucible at 350 °C for 30 min. This initial heating phase was performed to dehydrate the samples and remove CO_2_ and was then followed by melting at 1150 °C at a heating rate of 10 °C/min and held at this temperature for 90 min. The molten glass was then quenched between two stainless steel plates at room temperature or on stainless steel plates that had been kept at −20 °C for 4 h prior to casting. Once the glass had cooled it was broken up and ground using a Retsch PM100 milling machine and the ground glass samples were sieved to obtain particles in the range of 63–125 µm.

### 2.2. Manufacture of Solid and Porous PBG Microspheres

Microspheres were processed using a flame spheroidisation method that utilised a thermal spray gun (MK74, Metallisation Ltd., Dudley, UK). For solid microspheres, ground glass particles in a size range between 63–125 µm were fed into an oxy-acetylene flame at a gas ratio of 3:3. Calcium carbonate (CaCO_3_) was used as the porogen for porous microsphere manufacture. CaCO_3_ was sieved into a size range of 45–63 µm and mixed homogenously with the ground glass particles using a bench top vortex mixer in a 1:3 ratio. The glass and porogen were then fed into the flame and collected in the same manner as the solid microspheres. Post manufacture, the microspheres were cooled and collected before being seized at 63 µm. The microspheres were then submerged in 5M acetic acid for 120 s and then further washed in 2 L deionised water for 5 min. The solution was then filtered using a funnel and filter paper and dried overnight at 50 °C.

### 2.3. Scanning Electron Microscopy (SEM)

Morphological analysis of microspheres manufactured from the phosphate glasses was performed using a JSM-6490LV (JEOL, Peabody, MA, USA). Representative samples of the microspheres were mounted on carbon tabs attached to aluminium stubs and sputter coated with ~15nm of platinum under an argon atmosphere.

In order to examine the internal structures of the microspheres, a sample was embedded in cold set epoxy resin. The resin blocks were polished using SiC paper followed by a polishing cloth embedded with diamond paste down to 1 µM whilst using Industrial methylated spirit (IMS, Sigma Aldrich, UK) as a lubricating medium. The resin block was placed in an ultrasonic bath of IMS for 5 min and left to dry. The samples were then splutter coated with 15 nm carbon.

### 2.4. Mercury Porosimetry

The porosity of the microspheres was investigated using mercury intrusion porosimetry (Micromeritics Autopore IV 9500). A 5 cc powder penetrometer (Micrometrics) with a 1 cc intrusion volume was used for all samples. An empty penetrometer test was also conducted as a blank before running the samples.

### 2.5. Powder X-ray Diffraction (XRD)

A Bruker D8 Advanced X-ray diffractometer (Bruker-AXS, Karlsruhe, Germany) was used to determine the amorphous nature of the ground phosphate glass samples at room temperature using a Ni-filtered Cu-K_α_ radiation source. Data points were obtained every 0.02° from 10–70° over a 10 min period. The resulting data was analysed using DIFFRAC.EVA software (DIFFRAC-plus suite, Bruker-AXS) to identify phases through a database of known peaks from the International Centre for Diffraction Data (ICDD) database 2005. When studying porous microspheres following SBF immersion, data points were obtained every 0.01° from 5–70° over a 4 h period with a step size of 2.2 s.

### 2.6. Energy Dispersive X-ray Spectroscopy (EDX)

Compositional analysis was performed on ground glass samples and on solid and porous microspheres. Samples were mounted on carbon tabs attached to aluminium stubs and sputter coated with carbon using Q150T Turbo-Pumped Sputter Carbon Coater (Quorum, Lewes, UK). An Oxford Instruments INCA EDX system fitted with a Si-Li crystal detector was attached to the JSM-6490LV SEM and operated with an accelerating voltage of 15 kV and a working distance of 10 mm for EDX analysis.

### 2.7. Nuclear Magnetic Resonance

Quantitative 1D ^31^P MAS-NMR spectra were recorded at room temperature on a Varian Chemagnetics Infinity Plus spectrometer at a Larmor frequency of 121.47 MHz, using a 4 mm zirconia MAS probe spinning at approximately 10 kHz. The ^31^P 90° pulse duration was 3.5 μs, the spectral width was 100 kHz and the acquisition time was 10.24 ms. Chemical shifts are quoted relative to 85% H_3_PO_4_ in H_2_O, using Na_4_P_2_O_7_·10H_2_O as a secondary reference. The resulting spectra were deconvoluted into a series of Gaussian lineshapes, which were integrated to quantify the relative proportions of the Q^n^ sites. Deconvolution and fitting were achieved using a routine written in MATLAB and included 1st and 2nd order sidebands.

### 2.8. Ion Release and pH Studies

Ion release profiles of the glass formulations were determined by immersing 400 mg glass microspheres in 40 mL of ultrapure (Milli-Q) water at 37 °C. The dissolution medium at each time point (3, 7, 14, 21 and 28 days) was extracted via filtration and replaced. The concentration of sodium, phosphorus, calcium and magnesium ion release was analysed using inductively coupled plasma mass spectrometry (ICP-MS, Thermo-Fisher iCAP-Q model). The pH of the solutions was measured at each time point using a pH electrode InLab Pure Pro-ISM (Mettler Toledo, Royston, UK).

### 2.9. Bioactivity Studies

Simulated body fluid (SBF) was prepared according to ISO 23317:2014. The apatite-forming-ability of the bioactive glass particles within this study was performed using the TCO4 method. A total of 75 mg of phosphate-based glass microspheres and particles of 45S5 Bioglass were immersed in 50 mL SBF within Falcon tubes. The tubes were placed on an orbital shaker and agitated at 120 rpm whilst in a 37 °C incubator. The samples were incubated for different time points: 3, 7, 14, 21 and 28 days. At the end of each time period, the sample was removed from the solution by filtration using 20 µM filter paper. The particles were washed with DI water and then subjected to post-immersion characterisation testing involving SEM, XDR and EDX.

### 2.10. Microspheres Sterilisation and Preparation of Conditioned Media for Cell Study

Sterilisation of microspheres was performed throughout two washes of 15 min with ethanol 100% followed by complete evaporation at room temperature overnight in sterile conditions. For the preparation of conditioned medium containing microsphere ion extracts, 100 mg/mL of sterile microspheres were incubated in standard cell culture medium (SM) (DMEM supplemented with 10% foetal calf serum, 2% antibiotics-antimycotics, 1% L-Glutamine, 1% of non-essential amino acids, 2% HEPES and 0.015 *w*/*v* % ascorbic acid, Thermofisher, Cramlington, UK) at 37 °C and 5% CO_2_ for 48 h. The conditioned media containing microsphere ion extracts were collected and replaced with an equal volume of fresh medium every 48 h. Before being administrated to the cells, the solutions were filtered throughout 0.22 μm syringe filters to remove any debris or precipitate.

### 2.11. Indirect Cell Culture Study

In order to determine the cytocompatibility of the microspheres, an indirect culture method was performed by feeding the human osteoblast-derived cell line (MG63) (obtained from European collection of cell cultures—ECACC) with porous microsphere-conditioned medium to evaluate the biological response due to the different composition and dissolution products released by the microspheres over time. MG63s were seeded at a density of 10,000 cells/cm^2^ in 300 μL of standard cell culture medium in 48-well plates. Then, 48 h after seeding, the cells were washed with PBS and 300 μL of the appropriate conditioned media was added. Cells cultured with either unconditioned standard medium (+ve), or standard cell culture medium supplemented 5% DMSO (−ve) were used as controls. All media was refreshed every 48 h. Two independent biological replicates were performed with 3 experimental replicates for each condition.

### 2.12. Cell Metabolic Activity 

Cell metabolic activity of MG63 cells was evaluated at Days 2 and 7 using an Alamar Blue assay. A total of 300 μL of Alamar Blue solution (1:9 Alamar blue:Hanks Balanced Salt Solution) was added to each well and incubated for 80 min at 37 °C and 5% CO_2_ followed by a further 10 min on a shaker at 150 rpm. For each condition, three aliquots of 100 μL were transferred to a 96-well plate. A FLx800 fluorescence microplate reader (BioTek Instruments Inc., Winooski, VT, USA) was used to measure fluorescence at 560 nm excitation and 590 nm emission wavelengths. 

### 2.13. Alkaline Phosphatase (ALP) Activity

At day 7, the cells were washed three times with warm (37 °C) PBS and immersed in 1 mL of deionised water. The samples were freeze-thawed three times to lyse the cells and release nuclear content. An ALP assay (Randox, London, UK) was used to measure ALP activity in MG63 cells. Three aliquots of 50 μL of cell lysate were transferred to a 96-well plate and topped with 50 μL of ALP substrate (*p*-nitrophenyl phosphate 10 mM in diethanolamine buffer 1 mM at pH 9.8, with MgCl_2_ 0.5 mM). The plates were shaken gently at 150 rpm for 5 min on a plate shaker, and absorbance was measured at 5 min intervals until values no longer changed at a wavelength of 405 and 620 nm using an ELx800 microplate colorimeter (BioTek Instruments).

### 2.14. DNA Content Assay

Lysed samples used to measure ALP activity were thoroughly mixed using a vortex for 30–60 s and 100 μL of each sample was aliquoted into a 96-well plate. Hoechst 33258 stain was prepared (1 mg of BisBenzimide stain dissolved in 1 mL of deionised water and diluted to 1:50 in TNE buffer) and DNA standards were prepared using calf thymus DNA (Sigma, Welwyn Garden City, UK) and TNE buffer (10 mM Tris, 2M NaCl, and 1 mM EDTA in deionised water, adjusted to pH 7.4) as a diluent, to generate a standard curve for DNA concentrations. A total of 100 μL of Hoechst 33258 stain was added to each well and mixed on plate shaker for 5 min at 150 rpm. A FLx800 plate reader (BioTek Instruments) was used to read the plate at an excitation wavelength of 360 nm and an emission wavelength of 460 nm.

### 2.15. Direct Cell Culture Study

For direct seeding of the cells onto the porous PBG microspheres, MG63 cells were seeded at a density of 10,000 cells/cm^2^ onto 10mg sterilised porous microspheres from each formulation into low-adherent 48-well plates previously coated with 1% (*w*/*v*) solution of poly(2-hydroxyethyl methacrylate) (poly-HEMA, Sigma-Aldrich, UK) and Ethanol 95% in 300 μL of standard cell culture medium. The cells were cultured for 7 days at 37 °C and 5% CO_2_. The media was refreshed every 48 h. Two independent biological replicates were performed with 3 experimental replicates for each condition.

### 2.16. Direct Microsphere Cell Imaging

Cell imaging was performed after 7 days of direct culture with porous microspheres on cells that were fixed with 4% paraformaldehyde in PBS. For Environmental Scanning Electron Microscopy (ESEM), the post-fixed cells were washed twice with distilled water and analysed using a FEI Quanta 650 ESEM microscope.

### 2.17. Statistical Analysis 

For the cell metabolic assay and ALP assay, two independent experiments were performed, and the results are shown as mean ± standard error of mean (unless otherwise stated). Statistical analysis was performed using the Prism software package (v.9.2.0, GraphPad Software, San Diego, CA, USA, www.graphpad.com accessed on 23 February 2021). Two-way analysis of variance was calculated followed by a Tukey’s multiple comparison test. The mean difference was considered to be significant at 0.05 and with a 95% confidence interval. 

## 3. Results

### 3.1. Morphology and Topography of Microspheres

SEM analysis confirmed that following processing and sieving a high yield of spherical solid (SMS) (Figure 1A) and 22 highly porous microspheres (PMS) (Figure 1B) of each glass composition were produced via the flame spheroidisation process. Figure 1B represents the yield and distribution of pore morphologies obtained for the porous microspheres from the four different PBG formulations investigated. Cross-sectional SEM images of resin embedded porous microspheres, of each formulation, revealed that the pores were not limited to the surface of the microspheres but also exhibited inner porosity with interconnected pores (see Figure 1C).

### 3.2. Porosity Analysis

Mercury porosimetry analysis of porous phosphate-glass microspheres is shown in Figure 2. The mean pore size diameter for the porous microspheres of each formulation was determined to be 47, 48, 42 and 38 µm for P40, P35, P32.5 and P30, respectively, and indicated that a lower phosphate content resulted in smaller pore diameter (see Figure 2A). Mercury porosimetry also revealed the presence of submicron pores in addition to the larger pores visible under SEM analysis. As seen in Figure 2B, P40 porous microspheres exhibited the greatest porosity level of 75%, which gradually decreased to 70% for P35 and P32.5 and to 68% for P30 porous microspheres.

### 3.3. XRD Analysis

XRD analysis on solid microspheres showed that each of the glass formulations, with the exception for P30, processing via flame spheroidisation resulted in the formation of spherical glass particles which remained amorphous, as depicted by the lack of any sharp crystalline peaks in Figure 3A. Although the P30 solid microspheres had the same characteristic broad peak at 2θ values of ~20–40° as the other formulations, several small crystalline peaks at 24°, 32° and 35° were also observed. These peaks corresponded to Sodium Calcium Phosphate (ICDD 00-003-0751) and Sodium Phosphate (ICDD 00-030-1232) (see Figure 3A). XRD of the porous microspheres of each formulation revealed the presence of sharp peaks at ~29° 2θ and at 36°, 39° and 43° 2θ values. These peaks were matched to CaCO_3_ according to powder diffraction file 01-072-1937 (ICDD database). The same Sodium Calcium Phosphate (ICDD 00-003-0751) and Sodium Phosphate (ICDD 00-030-1232) peaks that were present in the P30 solid microspheres were also evident in the P30 porous microspheres (see Figure 3B).

### 3.4. Compositional Analysis of the Microspheres

Elemental values for the starting glass, solid and porous microspheres were obtained using EDX analysis. As seen in Figure 4, modest variation of chemical composition between the glass and solid microspheres was observed in all four formulations following spheroidisation. Processing P32.5 and P30 glass into solid microspheres resulted in slight increases of 1.7 and 0.4 mol % of P_2_O_5_ as compared to the starting glass particles. However, processing each of the formulations into porous microspheres resulted in far more significant changes in chemical composition. The porous microspheres from all four compositions exhibited elevated CaO content ranging from ~2.9–7.1%. This change was most prominent in the porous P40 microspheres, followed by P32.5 and then by P35 and P30. As a result of this CaO increase, all compositions of porous microspheres had lower MgO and Na_2_O values compared to the starting glass formulations produced. A decrease in P_2_O_5_ content between ~0.5–3.9% lower than the starting glass composition was also observed (see Figure 4A). The most significant decrease in P_2_O_5_ content was seen for P40 PMS and this correlated with the greatest increase in CaO content observed.

### 3.5. 1D ^31^P MAS NMR

Quantitative 1D ^31^P magic angle spinning (MAS) solid state nuclear magnetic resonance (NMR) spectroscopy was used to determine the Q_n_ species present in the phosphate glasses produced and to explore any changes due to processing the glasses into solid and porous microspheres.

As seen in Figure 5, as the phosphate composition decreased going from P40 to P30 in the phosphate glass, there was a gradual decrease in structural connectivity and an increased average number of non-bridging oxygen atoms (NBOs). The proportion of Q^2^ species decreased from 50% to 18% to 7% in P40, P35 and P32.5 glass, respectively, and were not detected at all for the P30 glass. The P40 glass was found to be composed of 50% Q^2^ and 50% Q^1^ species. However, for the other glasses in the series a decrease in the proportion of Q^1^ species from 81% to 75% to 68% for P35, P32.5 and P30 glass, respectively, was also observed. Conversely, the proportion of Q^0^ species observed for the P35 glass was less than 1%, which increased to 18% in P32.5 glass and significantly increased to 32% for the P30 glass composition.

When the phosphate glass particles were processed into both solid and porous microspheres, further depolymerisation of the glass network occurred, although this trend was less pronounced than the effect observed from decreasing phosphate content (see Figure 5). For the P40 glass, an increase in Q^1^ species to 61%, at the expense of Q^2^ was observed, which reduced to 35% and 1% Q^0^ were detected when they were formed into solid microspheres. Interestingly, the formation of porous microspheres resulted in a further reduction of Q^2^ species, to 31%, and increase in Q^0^ to 6% was also observed. Similar depolymerisation of the glass network occurred when the P35, P32.5 and P30 glasses were processed into solid and porous microspheres, albeit to a lesser extent. Processing the starting glass materials into porous microspheres resulted in the most significant glass network depolymerisation in comparison to solid microspheres.

### 3.6. Ion Release Studies

Figure 6 shows the cumulative ion release profiles, calculated from measurements obtained via Inductively Coupled Plasma (ICP) analysis, for the porous microspheres following immersion in milli-Q water over 28 days. For each of the four porous microsphere formulations, the ions measured appeared to exhibit linear release rates over the measured time period. A clear trend in the release rate of each ion was seen across the microsphere series. P40 microspheres released the highest quantity of all four glass forming ions and as the phosphate content in the microspheres decreased, so did the release rate for all ions. Porous P40 microsphere released the highest Na^+^ (132 ppm), Mg^2+^ (71 ppm), Ca^2+^ (8 ppm) and P (286 ppm), whereas P30 released the lowest Mg^2+^ (22 ppm), Ca^2+^ (37 ppm) and P (40 ppm). For Na^+^, Mg^2+^ and P released from P35, P32.5 and P30 PMS, relatively similar ion release profiles were seen, whereas P40 microspheres released a significantly greater amount of these ions. For instance, P40 microspheres released more than 2.5 times more Na^+^ than the P35 microspheres, the second highest producer of Na^+^ ions, and approximately 4 times the amount of P ions.

Figure 6E shows the pH of the milli-Q water used to immerse the porous microspheres over the 28-day period. For the solution containing P40 porous microspheres, there was a small decrease in pH from ~7.8 at day 3 to ~7.0 by day 28. The pH of the solution at day 3 was higher, between 9.2 and 8.8, for the other three glass formulations, which gradually decreased for all formulations by day 7 and a clear trend was established with the lower phosphate containing glasses, revealing a higher pH of the milli-Q water.

The slightly alkaline pH of the microsphere-containing milli-Q water was attributed to any remaining residual porogen retained on or within the microspheres. As seen in Figure 3, the XRD traces revealed the presence of CaCO_3_ within porous microsphere but not for the solid microspheres post-processing. As seen in Figure 6E, lower pH values were observed for the higher phosphate containing microspheres, which was likely due to the release and breakdown of phosphate chains leading to more PO_4_ groups and a subsequent increase in PO_4_^3−^ ions into the solution [4]. In order to confirm the role that residual CaCO_3_ within porous microspheres was having on the pH of the solution, solid microspheres of the same formulation and size were also immersed in milli-Q water for 28 days. The pH of the solutions containing the solid microsphere followed the same trend but were at a slightly lower, more neutral pH (Figure 6F). This was most evident between days 3 and 7.

### 3.7. Bioactivity Studies in Simulated Body Fluid

Since porosity can facilitate enhanced gaseous and nutrient exchange and encapsulation of therapeutic compounds, the bioactivity of porous P40, P35, P32.5 and P30 microspheres was explored to assess their suitability for in vivo orthobiologic applications.

The XRD profiles of the four PBG porous microspheres and 45S5 Bioglass, which was used as a positive control, following 28 days of immersion in SBF are shown in Figure 7A. The XRD spectra for 45S5 were in good agreement with the results seen in other studies, with sharp peaks at 2θ values of 26° and 32° from day 7 onwards revealing hydroxycarbonate apatite (HCA) (ICSD 01-084-1998) formation [21]. The spectra obtained for P40 and P35 lacked the sharp peaks at 2θ values of 26° and 32° that indicate the formation of HCA on the microsphere surface. A prominent peak at 2θ value of 29° and smaller peaks at 36°, 39°, 47° and 48° correlated with peaks for CaCO_3_ (ICSD 00-024-0027) that had previously been identified using XRD on porous microspheres, prior to SBF immersion (Figure 3). Peaks at 29°, 36°, 39°, 47° and 48° were present at each time point. P32.5 and P30 spectra displayed peaks for CaCO_3_ at each time point and by 28 days peaks at 2θ values of 31° and 34° were also visible which were matched to DCPD (i.e., brushite) (ICSD 00-004-0740).

EDX analysis was also performed and used to calculate the Ca:P ratio (wt%) of the porous microspheres prior to and after 28 days of immersion in SBF. The porous microspheres from each of the formulations exhibited elevated Ca:P ratios in comparison to their parent glass formulations, which was attributed to the increase in CaO content from use of the CaCO_3_ as porogen. This was most pronounced in the P40 and P35 PMS (as seen in Figure 7B), where elevated CaO correlated with increased porosity (see Figure 2). Following immersion in SBF for 28 days, microspheres from each glass composition exhibited an increase in the Ca:P ratios, which were used to predict any potential phases present on the surface of the microspheres (see Figure 7B).

### 3.8. Indirect In Vitro Cell Culture Studies

In order to determine the cytocompatibility of the porous glass microspheres, an indirect cell culture method was performed which involved feeding microsphere-conditioned media to osteoblast-like cells (MG63s) to evaluate their biological response to the dissolution products released by each microsphere composition investigated, over time. Standard medium (SM) and SM containing 5% DMSO were included as positive and negative controls, respectively. Analysis of metabolic activity via the Alamar Blue assay showed a significant increase in cell response when treated using each of the four porous microsphere-conditioned media between days 2 to day 7. This increase in metabolic activity was also seen for cells treated with SM but cells treated with SM + 5% DMSO did not have an increase in metabolic activity from day 2 to day 7 (D2 vs. D7: *p* < 0.0001). No significant difference was detected between all formulations at day 2, apart from comparison to SM +5% DMSO (see Figure 8A). Although not statistically significant, a trend for increased metabolic activity for the lower phosphate containing microsphere formulations was also observed.

At day 7, the trend towards increasing metabolic activity with decreasing phosphate content in the microspheres became even more significant. Cells grown in P30-conditioned media had a significantly higher metabolic activity response to cells grown in all other conditions except for those in P32.5-conditioned media (vs. P40, P35, SM, SM+5% DMSO *p* < 0.0001). Cells grown in P40 media had a significantly lower cell response in comparison to P32.5 (*p* < 0.05) and P30 cells (*p* < 0.0001), but this was comparable to cells grown in P35-conditioned media and SM (Figure 8A).

Alkaline phosphatase (ALP) activity was measured as an early marker of osteogenic differentiation in MG63 after 7 days of indirect culture from the four phosphate-glass microsphere formulations. The ALP activity was normalised to the DNA content of the samples investigated. Cells grown in P40-conditioned media had significantly higher ALP activity compared to cells grown in the three other microsphere-conditioned media as well as standard media (vs. P35, P32.5 and P30: *p <* 0.001; vs. SM: *p* < 0.05). There was no statistically significant difference in ALP activity between cells grown in SM and those grown with P35-, P32.5- and P30-conditioned media (see Figure 8B).

### 3.9. Direct In Vitro Cell Culture Studies

Seeding of MG63 cells directly onto the microspheres was performed in order to evaluate the cellular response to direct contact with the four porous microsphere formulations and assess their suitability as biomaterials for bone repair and regeneration applications. The analysis of the metabolic activity of cells in direct physical contact with the microspheres showed that after 2 days of cell culture there was no statistically significant difference in cells grown on the four different microsphere formulations (Figure 9A).

Metabolic activity decreased slightly by day 7, with this response only being significantly lower for P32.5 microspheres. After 7 days of culture there was no statistically significant difference in metabolic activity measured between cells cultured on the four microsphere formulations. The metabolic activity of cells cultured on tissue culture plastic (TCP) as a positive control was significantly lower at day 2, and by day 7 was significantly higher (Figure 9A).

ALP was evaluated after 7 days of direct culture and was also normalised to DNA concentrations of each sample. ALP activity can be used as a measure of early osteoblastic differentiation as it is an early marker of cell matrix maturation. The ALP activity after 7 days of MG63 cell growth on all four microsphere formulations was significantly higher than those grown on the TCP control (*p* < 0.0001), although no statistically significant difference was detected between the formulations (Figure 9B).

MG63 cells that were directly cultured onto the four microsphere formulations were visualised using ESEM at day 7 to further assess whether the material surface would support cell growth. The cells were shown to adhere onto the microspheres from each of the four glass formulations (see Figure 10). The MG63 cells had spread and colonised the pore regions within the microspheres structure as clearly shown in Figure 10. The MG63 cells displayed lamellipodia and filopodia projections which were also seen bridging adjacent neighbouring microspheres as well as penetrating into the pores.

## 4. Discussion

Recently, studies have shown that bioactive materials consisting mainly of pyro- and ortho- phosphate species show highly favourable cytocompatibility responses [13]. These phosphate species have been shown to play a vital role in bone mineralisation as well as stimulating osteogenic differentiation, matrix gene expression and ALP activity in osteoblasts [14].

In this work, PBG formulations termed P40, P35, P32.5 and P30 were produced and EDX analysis confirmed that the elemental compositions of the starting glasses produced were within 1.5% error margin of the target values. This is commonly seen in other studies involving phosphate glasses where a 1–2 mol% error between the theoretical and empirically obtained values is usually observed [22]. The slight discrepancies between the expected and target values reported in other studies on phosphate-based glasses are often attributed to the hygroscopic nature of phosphate precursor salts used in the glass manufacturing process [23].

A high yield of both solid and porous microspheres, in the size range of 125–200 µm, was successfully processed from each composition using the flame spheroidisation process developed in our group. For the porous microspheres at each composition, pores of varying diameters were present throughout the surface and inner body of the microspheres. The pores were shown to be interconnected with the average pore diameter ranging from 38–48 µm between the formulations. Mercury porosimetry revealed the presence of submicron pores within the porous microspheres and determined that there was little variation in the porosity level between the four formulations.

XRD analysis confirmed that processing of the P40, P35 and P32.5 glass formulations into solid microspheres resulted in the retention of their amorphous nature. However, when processing into porous microspheres, crystalline peaks were observed which were attributed to residual porogen remnants within the microspheres which had not been fully removed during the wash step. Conversely, crystalline regions were also detected for the P30 solid and porous microspheres by XRD (see Figure 3B).

Structural analyses revealed that as the phosphate content of the glass particles and the microspheres decreased, a gradual decrease in structural connectivity (Q^2^ to Q^1^ to Q^0^) and an increased average number of non-bridging oxygen atoms was also observed. The ^31^P MAS-NMR data showed an absence of Q^0^ species for P40 glass and a considerable increase from less than 1% for P35 glass up to 18% and 32% for P32.5 and P30 glass, respectively. The increase in Q^0^ as phosphate content decreased was accompanied by a subsequent decrease in Q^2^ and Q^1^ species (see Figure 5). These changes were attributed to the higher proportion of modifier oxides present in the lower phosphate content glasses, which indicated that more non-bridging oxygen atoms were needed to charge balance the cations. This led to a greater number of oxygen atoms that were therefore unable to covalently bond with adjacent phosphate tetrahedra leading to depolymerisation of the glass network [24].

Processing the glasses produced into porous microspheres resulted in a more significant change in the Q^n^ species than processing into solid microspheres. EDX analysis revealed that the addition of CaCO_3_ (which was used to induce porosity in the microspheres) resulted in an increase in CaO content compared to the initial glass sample. This increase caused a subsequent decrease in the proportion of P_2_O_5_, MgO and Na_2_O (see Figure 4). The additional calcium cations incorporated into the glass structure introduced an excess positive charge that resulted in an increase in the quantity of non-bridging oxygen atoms to neutralise the charge balance. In order to achieve a greater proportion of non-bridging oxygen atoms, network depolymerisation must occur and is seen with the increased proportion of Q^0^ and a decrease primarily in Q^2^ but also Q^1^ species [25].

A high proportion of orthophosphate (Q^0^) and pyrophosphate (Q^1^) species were noted for the P32.5 and P30 glasses, which were most likely responsible for the high crystallisation tendency of these glasses during manufacture (especially P30) [12]. The tendency for a glass to crystallise is closely associated with the viscosity of the melt, with lower viscosities facilitating the arrangement of the components into an ordered crystalline structure more easily. The disrupted structure created by the short phosphate units meant that the P30 and P32.5 glasses had a lower viscosity of the melt and therefore a higher tendency to crystallise [25]. This was the likely reason that crystalline peaks were observed for P30 SMS and PMS following processing but were not seen in the other formulations. Nevertheless, it should be noted that porous microspheres were still successfully produced from both these formulations.

The porous P40 microspheres released a significantly greater amount of all the ions in comparison to the other three porous microsphere formulations. Microspheres from each of the four porous formulations exhibited linear ion release profiles and as the phosphate composition decreased from P40 to P30, this was accompanied by a subsequent decrease in ion release (see Figure 6). The quantity and rate of ion release from P40 PMS correlated well with previous studies which had established that porous microspheres had increased ion release rates and mass loss during degradation in comparison to solid microspheres of the same formulation, as a result of an increased surface area [20]. It is well-established that the dissolution behaviour of phosphate glasses depends on the phosphate anions and the associated metal ions that constitute the glass structure [26]. The P35, P32.5 and P30 formulations are considered as invert glasses as their properties are controlled by the interactions of cations with the phosphate groups rather than the P_2_O_5_ network and the entanglement of phosphate chains [6]. P32.5 and P30 microspheres released the lowest amount of all the ions and in comparable amounts, including Ca^2+^ of which they contained a greater amount in comparison to the other formulations. Calcium ions within phosphate glasses are known to have strong complexing abilities with linear polyphosphates forming cross-linking interactions which strengthen the glass network [27]. A more compact phosphate glass network, due to the presence of ortho and pyrophosphate groups and increased cross-linking, most likely contributed to the lower solubility and subsequent ion release profiles due to the inability of water molecules to easily penetrate and hydrate the structure [28]. Although there was variation in the quantities of ions released, all four porous microspheres formulations released calcium, phosphate, magnesium and sodium ions throughout the duration of the study. Continued, localised ion release from the microspheres could deliver therapeutic ions in a site-specific manner, which would not only optimise their therapeutic efficacy but minimise any off-target effects [29]. The role of ions, such as calcium, phosphate and magnesium, within the bone regeneration process has been established with them primarily acting as enzyme co-factors. They are therefore able to stimulate various signalling pathways and their controlled release can influence stem cell differentiation down specific lineages and be harnessed to increase osteogenesis [30]. The conversion of mesenchymal stem cells (MSCs) down the osteogenic lineage into osteoblasts is a vital process to facilitate bone mineralisation at sites of osseous tissue damage during bone remodelling and repair [31]. ICP analysis was unable to determine the specific phosphate species released by the microspheres. However, it has been reported that some of the phosphate anions released into solution are the same structural units present in the glass. Phosphate chains released from the glass surfaces then dissolve (via hydrolysis) in aqueous conditions to smaller phosphate units [32].

XRD analysis was performed to identify and differentiate crystalline phases present on the glass microspheres following immersion in SBF. By day 28, P30 and P32.5 microspheres exhibited peaks that corresponded to DCPD, whilst P40 and P35 porous microspheres only exhibited CaCO_3_ peaks that were present in all the compositions. The Ca:P ratios at day 28 for P32.5 (0.68) and P30 (0.84) porous microspheres were within the range suggested for precipitated amorphous calcium phosphate (ACP) (0.67–1.5) [33]. It has been reported that precipitated ACP formed on calcium phosphate biomaterials acts as a precursor phase towards the eventual formation of HA [34]. Dicalcium phosphate dihydrate (DCPD) and other calcium orthophosphates, such as octacalcium phosphate (OCP) are also known to form prior to the precipitation of HA crystals on phosphate-based bioactive glasses [35]. The lack of apatite-like depositions on metaphosphate glasses has been reported in other studies. Kasuga et al. performed NMR analysis of ternary phosphate glasses in the system *x*CaO·(90 − *x*)P_2_O_5_·10TiO_2_ (*x* = 45~60) [36]. They reported that 60CaO·30P_2_O_5_·10TiO_2_ glass contained orthophosphate and pyrophosphate groups, whereas glasses with a higher P_2_O_5_ content consisted predominantly of metaphosphate groups. They also studied the apatite-forming ability and stated that a bonelike apatite phase formed on the surface of the 60CaO·30P_2_O_5_·10TiO_2_ glass particles following 7 days immersion in SBF. Additionally, they reported an apatite-like phase was not found to have formed on the metaphosphate glasses [37]. However, the P30 and P32.5 glass formulations developed which contained high proportions of orthophosphates and pyrophosphates did exhibit greater bioactivity, which could lead to more favourable environments to promote bone repair and regeneration than other glasses containing higher phosphate species.

From the indirect cell culture studies performed (using microsphere-degradation- conditioned media from each of the four formulations) to assess cellular responses, it was demonstrated that the ion release products from the microspheres were cytocompatible. The cells grown in all four microsphere-conditioned media had either comparable or significantly increased metabolic activity in comparison to the cells grown in SM (see Figure 8A). At day 7, a trend was observed where increasing metabolic activity correlated with decreasing phosphate content in the microspheres. The cells grown in P30-conditioned media revealed a significantly higher metabolic activity response to the cells grown in all other conditions except for those in P32.5-conditioned media. Ion release studies (see Figure 6) showed that the release rate of all ions decreased as the phosphate content in the microspheres decreased. As highlighted above, the ^31^P MAS NMR revealed that P30 microspheres were formed exclusively from ortho- and pyrophosphate species and that P32.5 were also comprised of a greater proportion of these species in comparison to P35 and P40 microspheres. It is suggested that the concentration of ions and the specific phosphate species released from the microspheres resulted in the increased metabolic activity for the MG63 cell line. As shown in Figure 6E, the ion release products from the four porous microsphere formulations did not result in radical changes or fluctuations to the pH of the solution. Therefore, the profiles observed most likely created a stable and highly favourable microenvironment for cell growth and proliferation.

After 7 days of indirect culture from the four phosphate-glass microsphere formulations, the MG63s grown in P40-conditioned media revealed significantly higher ALP activity in comparison to the three other microsphere formulations (see Figure 8B). As seen in Figure 6, the P40 microspheres degraded faster and hence released greater ion quantities. In a study by Gupta et al., hMSCs exposed to solid microspheres, with the same P40 glass formulation also resulted in higher ALP activity compared to cells cultured in the slower-degrading borosilicate (1.7CaO·7.11Na_2_O·78.6SiO_2_·9.5B_2_O_3_·3.1Al_2_O_3_) and P45 (45P_2_O_5_·16CaO·24MgO·11Na_2_O·4Fe_2_O_3_ mol%) microsphere-conditioned media [16]. The correlation between increased ALP activity for the faster degrading microsphere formulation, in comparison to the slower degrading formulations, was also seen in the present study. Intracellular Ca^2+^ has been shown to enhance glutamate secretion inside cells, which has been proposed as one mechanism for promoting the osteogenic fate of osteoprogenitor cells [38]. The Ca^2+^ and other ions released from the P40 microspheres may have led to a greater osteogenic effect in comparison to the other microsphere formulations investigated. Conversely, Gupta et al. showed hMSCs exposed to P40 solid microsphere media also resulted in cells revealing a higher metabolic activity in comparison to the slower degrading microsphere formulations, which was not seen in this study. In the Gupta et al. study, the other glass formulations were a borosilicate glass and iron-containing phosphate glass, whereas the glasses used in this study were closer in their composition. It is vital to assess the effect that degradation products of the microspheres have on the cell response as this greatly influences the success of a material for bone repair applications.

The direct cell culture method exposes cells to both the effects of the microsphere dissolution products as well as to the physical environment via contact with the materials themselves. All four porous microsphere formulations showed cell growth over the 7 days of culture tested (see Figure 9A). In contrast to the indirect study, there was no significant difference in the metabolic activity of cells regardless of which microsphere formulation they were grown on. A study by De Melo et al. explored how tailoring the phosphate species within phosphate-based glasses effected the adhesion of hMSCs to the glass surface. As the phosphate content decreased from 40 to 30 mol% in the glass series (40 − *x)*P_2_O_5_·(16 + *x*)CaO·24MgO·20Na_2_O (*x* = 0, 5 and 10 mol%) depolymerisation of the glass network resulted in the formation of a greater abundance of ortho- and pyrophosphate species at the expense of metaphosphate species. This decrease in phosphate content in this glass series correlated with an increase in the number of hMSCs that adhered to the polished surface of the flat glass discs and greater metabolic activity of the cells after 24 h. This effect was not seen in the present study, where MG63 cells were used and cultured over a longer time period on phosphate glass but with a different morphology and geometry. In addition to the morphological difference between porous microspheres and glass discs, processing the glass into porous microspheres results in compositional changes (i.e increased CaO content) which may affect cell adhesion and metabolic activity.

Previous studies involving direct cell culture with phosphate-based glass microspheres showed that cells were capable of attaching to and proliferating on and between adjacent microspheres alongside the colonisation of the inner pore regions [20]. This was also observed in the present study, further indicating the suitability of the porous microspheres as substrates for cell attachment, growth and proliferation (Figure 10). Biomaterials that facilitate bone repair and regeneration would ideally be both osteoinductive (induce stem cell differentiation down the bone lineage) and osteoconductive (provide optimum conditions for bone growth). Additional desirable characteristics that significantly enhance the regenerative potential of the material are mechanical stability and porosity [39]. An interconnected 3D network of pores has been shown to facilitate cell attachment and allow for the infiltration of osteoblastic and vasculature of cells that support matrix deposition and the ingrowth of new bone [40]. Microporosity has been attributed to promoting bioactivity and protein interaction, whereas larger pores, ranging from 1–100 µm, support cell adhesion, proliferation and migration [41]. Porosity is a vital feature for a biomaterial to facilitate tissue repair as it allows for differentiated cells to make abundant and specialised extracellular matrix and allows bioactive molecules from the material to have access to the cells [42].

The essential and synergistic role of phosphate anions in combination with calcium and magnesium cations for the development of new bone tissue is well established [43]. Ortho- and pyrophosphates can directly influence the natural cycle of bone formation and resorption and are utilised to increase the generation of new bone tissue and stimulate HA crystallisation. Orthophosphates are vital for the mineralisation of collagen fibres formed from osteoblasts within the remodelling process [44]. Inorganic pyrophosphates can modulate biomineralisation by acting as both an inhibitor for mineralisation but also as a source of orthophosphates following their hydrolysis by ALP. Furthermore, pyrophosphates are capable of stimulating extracellular matrix gene expression, ALP activity and differentiation of pre-osteoblasts [14].

The phosphate-based glass formulations produced herein have a huge potential for hard tissue engineering applications due to their ability to influence biomineralisation through their release of specific ions in a controlled manner [45]. Microspheres that release ortho- and pyrophosphate species may provide enzymatically controlled levels of the inorganic components required for mineralisation [15]. Biomaterials such as these can be tailored to induce specific biological responses as opposed to materials that are designed solely to mimic a component of a biological structure [46]. The P32.5 and P30 porous microspheres used in this study were designed to release ortho- and pyrophosphate anions in addition to calcium, magnesium and sodium cations to create a favourable microenvironment for the migration, proliferation and differentiation of osteoblast and pre-osteoblast cells. They act not only as vehicles to deliver ions but provide a surface structure to improve osteointegration of new bone tissue, whilst limiting inflammatory or fibrotic responses following in vivo delivery. The microsphere size and morphology also facilitates their delivery via minimally invasive (injection) techniques and allows them to be utilised for orthobiologic applications [47]. Ideally, the microspheres would release ions that stimulate osteogenesis, support cell growth upon their surface and degrade at a suitable rate for new bone tissue formation [48].

Surface structures and geometries have also been shown to directly influence cellular responses including adhesion, proliferation and osteogenic differentiation [49]. In vivo studies have indicated that collagen deposition and new bone formation occur preferentially at concavities on biomaterial surfaces, further signifying the potential for porous microspheres to influence osteogenic differentiation over extended culture periods [50]. It may be that these concavities resulted in the increased ALP activity of cells grown directly on the microspheres in comparison to those on TCP (see Figure 9B). The increased ALP activity seen for cells cultured on the porous microspheres suggests that the microspheres provided a more favourable surface for influencing osteoblast cell differentiation as compared to the TCP control. A possible upregulation in ALP activity of the cells directly grown on the microspheres may explain the slight reduction in metabolic activity seen at day 7 in comparison to day 2. This was perhaps due to the upregulation in ALP activity being coupled with a downregulation of cell proliferation as maturation of the extracellular matrix occurred [51].

This work demonstrated that highly porous phosphate glass microspheres containing only ortho and pyrophosphate species are highly promising materials for bone repair and regeneration. This study shows for the first time that these novel ortho and pyrophosphate rich glass formulations could be processed into solid and porous microspheres. Microsphere porosity is desirable as it increases surface area which can allow for greater cell attachment, increased release of therapeutic ions and can be exploited for encapsulation and release of therapeutic cargo for orthobiologic applications. Previous in vivo studies have shown that the incorporation of autologous bone marrow cells within P40 porous microspheres and other biologics, such as drugs, proteins and growth factors, could also be loaded within the microspheres. The simple and rapid manufacturing process developed can produce high yields of porous microspheres with fully interconnected porosity using the flame spheroidisation method. The porous microsphere formulations investigated in this study warrant further investigation to evaluate their effect on osteoconduction and osteoinduction and the formation and maturation of new bone tissue in vivo.

## 5. Conclusions

This study showed that solid (SMS) and porous microspheres (PMS) from four quaternary PBGs in the series xP_2_O_5_·(56 − *x*)CaO·24MgO·20Na_2_O (mol%), where *x* is 40, 35, 32.5 or 30, can be prepared via a flame spheroidisation process. The porous microspheres showed interconnected porosity with an average pore size of 38–48 µm. The use of CaCO_3_ for PMS manufacture altered the chemical composition of the microspheres compared to the starting glass and increased network depolymerisation. Alterations to microsphere formulations and geometry can be used to tailor the ortho and pyrophosphate species present in the final product.

Tailoring the PMS formulation through increases in pyro- and orthophosphate species was shown to significantly reduce the ion release rates from the PMS. The P32.5 and P30 PMS containing a high proportion of ortho and pyrophosphate species demonstrated favourable bioactivity responses despite the reduced ion release rates. The localised release of cations and anionic ortho and pyrophosphate species from the PMS represent a promising resource and strategy for bone repair and regeneration.

The In vitro cell culture studies confirmed the cytocompatibility of all four PMS formulations investigated and showed that they provided favourable surfaces to facilitate cell adhesion and proliferation.

Although no statistically significant differences between the formulations were observed from direct cell culture studies, the cells treated with P30-conditioned media revealed higher metabolic activity from the indirect studies, whilst those treated with P40-conditioned media demonstrated elevated ALP activity. This suggests that a combination of microsphere formulations could be developed to elicit the desired responses. However, further In vitro and in vivo studies would be required to establish whether a specific microsphere formulation or a combination of different formulations would deliver the desired properties for facilitating enhanced bone regeneration.

## Figures and Tables

**Figure 1 bioengineering-09-00611-f001:**
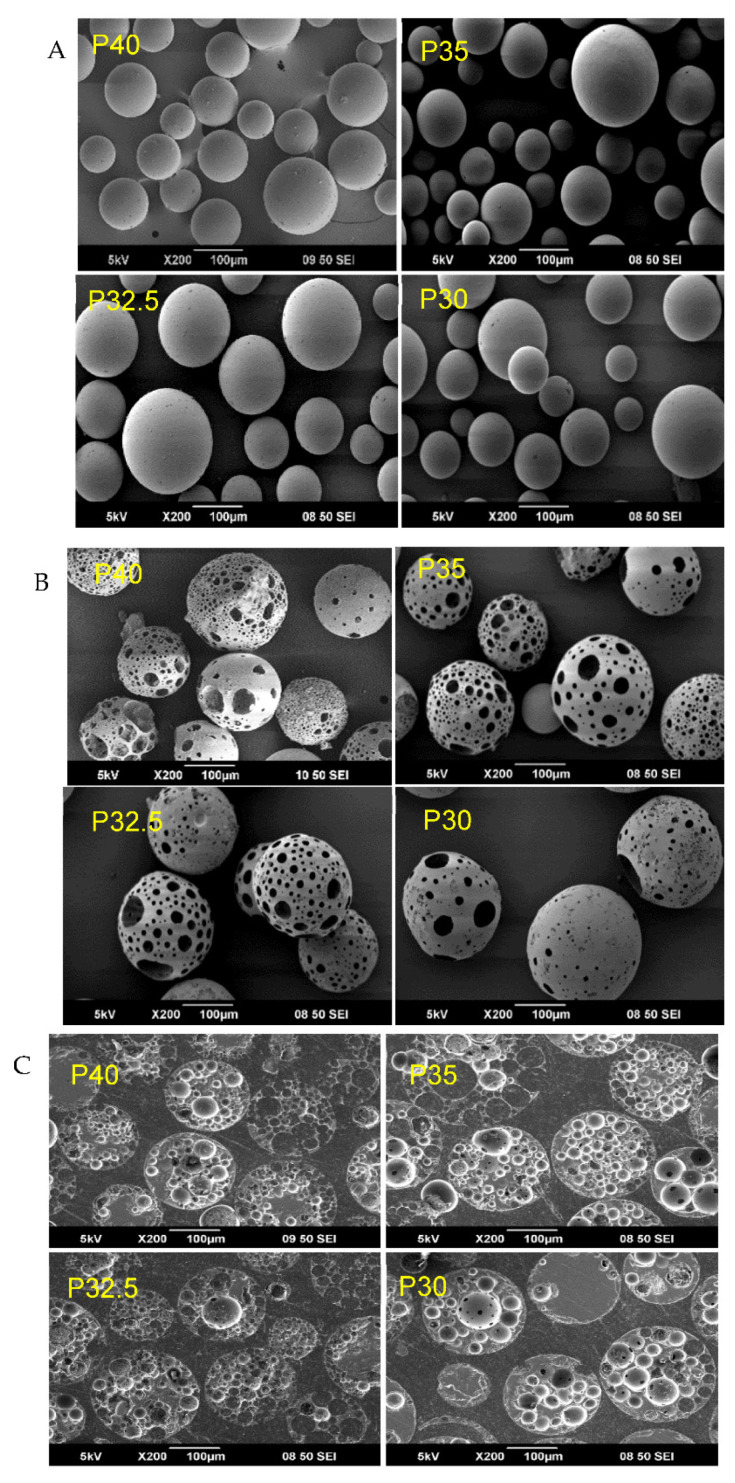
(**A**) SEM images of P40, P35, P32.5 and P30 solid microspheres, (**B**) SEM images of P40, P35, P32.5 and P30 porous microspheres and (**C**) SEM images of resin embedded P40, P35, P32.5 and P30 porous microspheres revealing their cross-section and showing the interconnected porosity.

**Figure 2 bioengineering-09-00611-f002:**
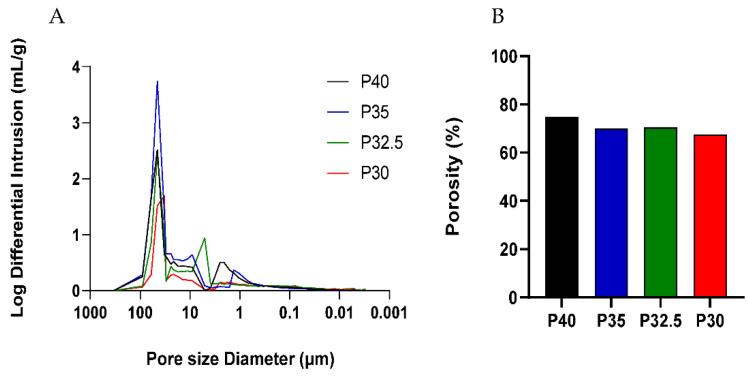
(**A**) Log differential intrusion curves of porous PBG microspheres highlighting the range of pore sizes achieved, and (**B**) Porosity (%) of microspheres obtained from the four PBG formulations.

**Figure 3 bioengineering-09-00611-f003:**
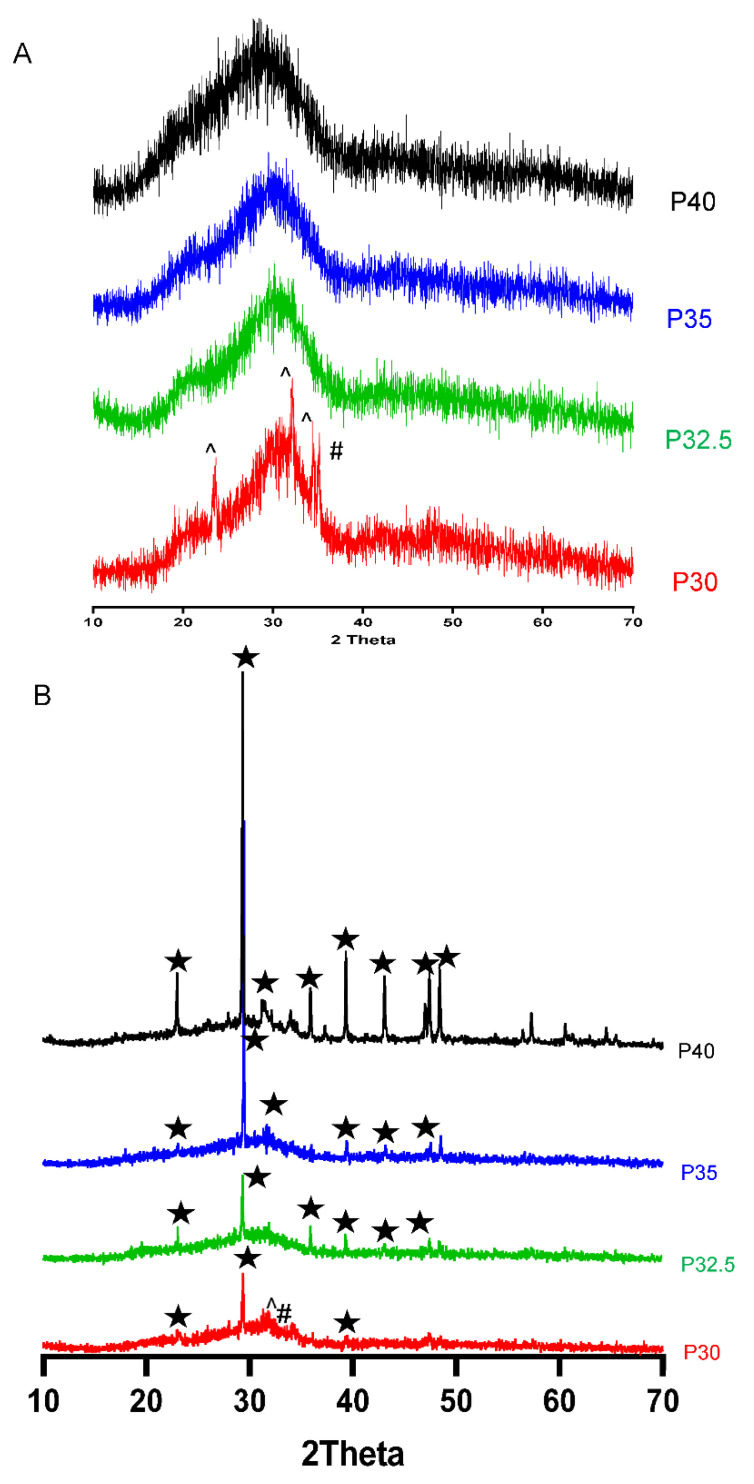
XRD spectra of (**A**) P40 (black), P35 (blue), P32.5 (green) and P30 (red) solid microspheres and (**B**) P40 (black), P35 (blue), P32.5 PMs (green) and P30 (red) porous microspheres. The crystalline peaks matched for Calcium Carbonate (★) (ICDD 01-072-1937), Sodium Calcium Phosphate (^) (ICDD 00-003-0751) and Sodium Phosphate (#) (ICDD 00-030-1232).

**Figure 4 bioengineering-09-00611-f004:**
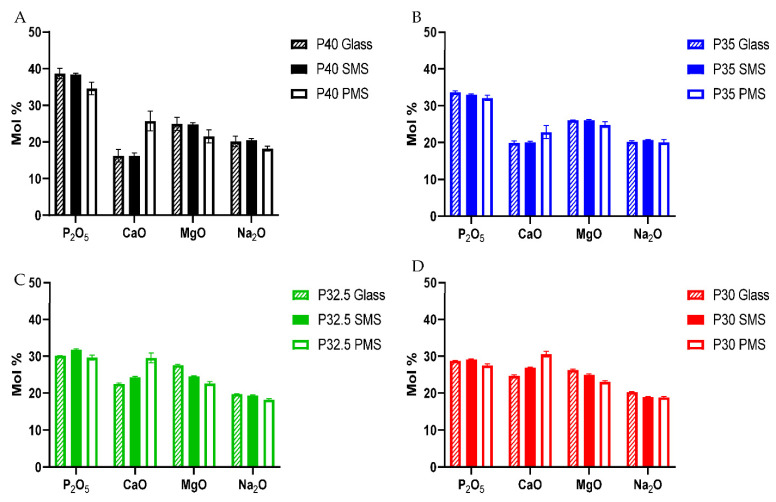
Chemical composition of the starting glass, solid and porous microspheres ascertained via EDX analysis. (**A**) P40, (**B**) P35, (**C**) P32.5 and (**D**) P30.

**Figure 5 bioengineering-09-00611-f005:**
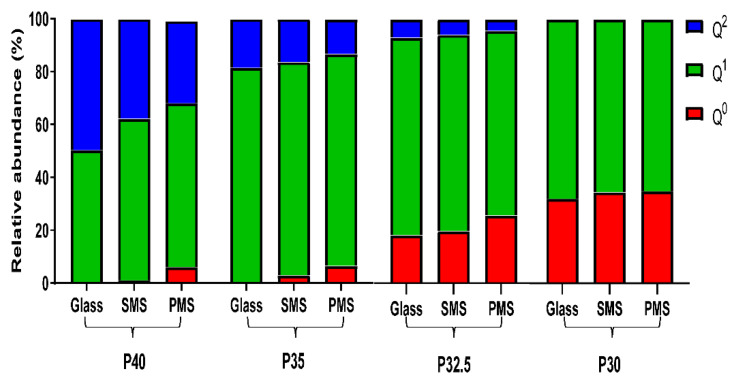
Relative abundance of Q^2^, Q^1^ and Q^0^ species for P40, P35, P32.5 and P30 glass, solid (SMS) and porous microspheres (PMS) obtained using 1D 31P MAS-NMR.

**Figure 6 bioengineering-09-00611-f006:**
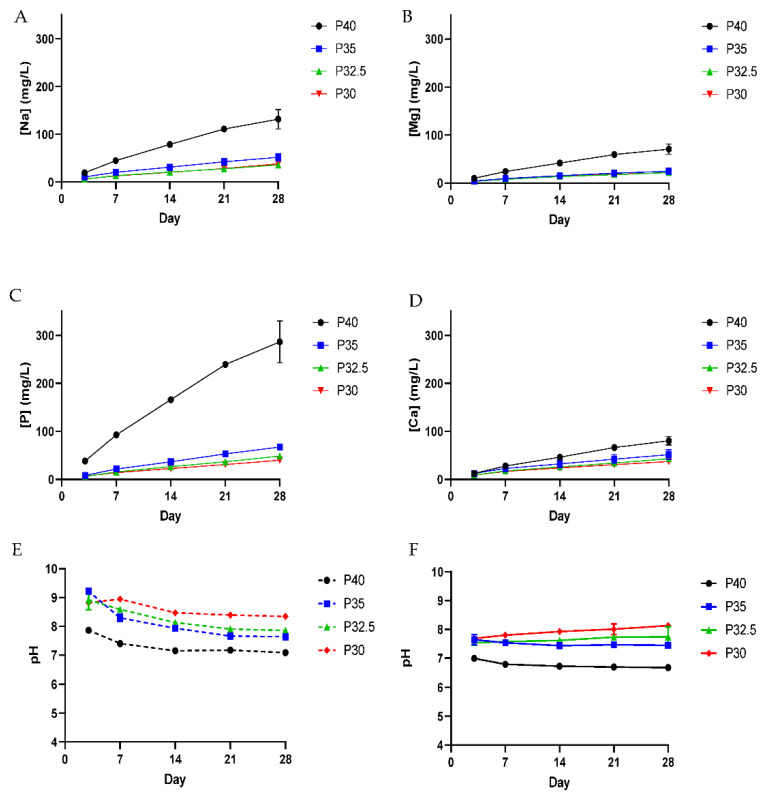
Cumulative ion release profile of (**A**) sodium, (**B**) Magnesium, (**C**) phosphorous and (**D**) calcium from P40 (black), P35 (blue), P32.5 PMs (green) and P30 (red) porous microspheres in milli-Q water over 28 days at 37 °C. (**E**) pH change of milli-Q water as a function of immersion time for porous and (**F**) solid microspheres.

**Figure 7 bioengineering-09-00611-f007:**
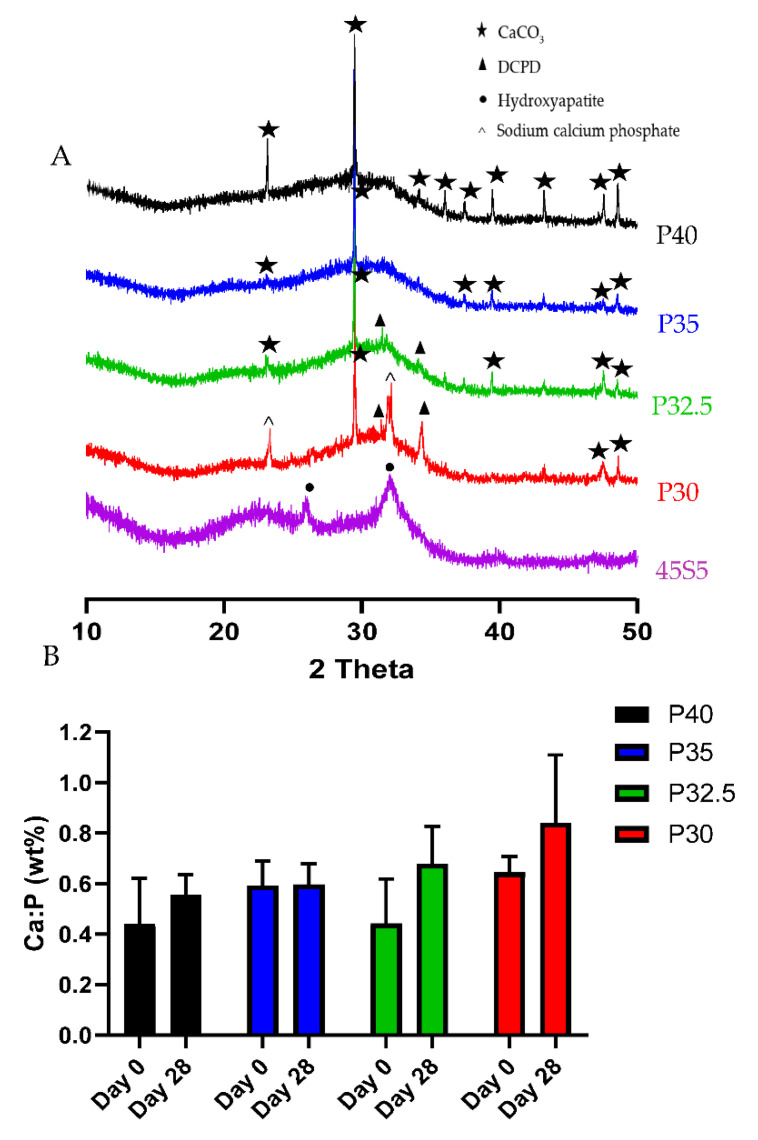
(**A**) Powder XRD patterns for P40 (black), P35 (blue), P32.5 (green), P30 (red) porous microspheres and 45S5 (purple) particles following 28 days of immersion in SBF. (**B**) Ca:P ratio, determined by EDX, for porous microspheres from each composition prior to SBF immersion and after 28 days of immersion.

**Figure 8 bioengineering-09-00611-f008:**
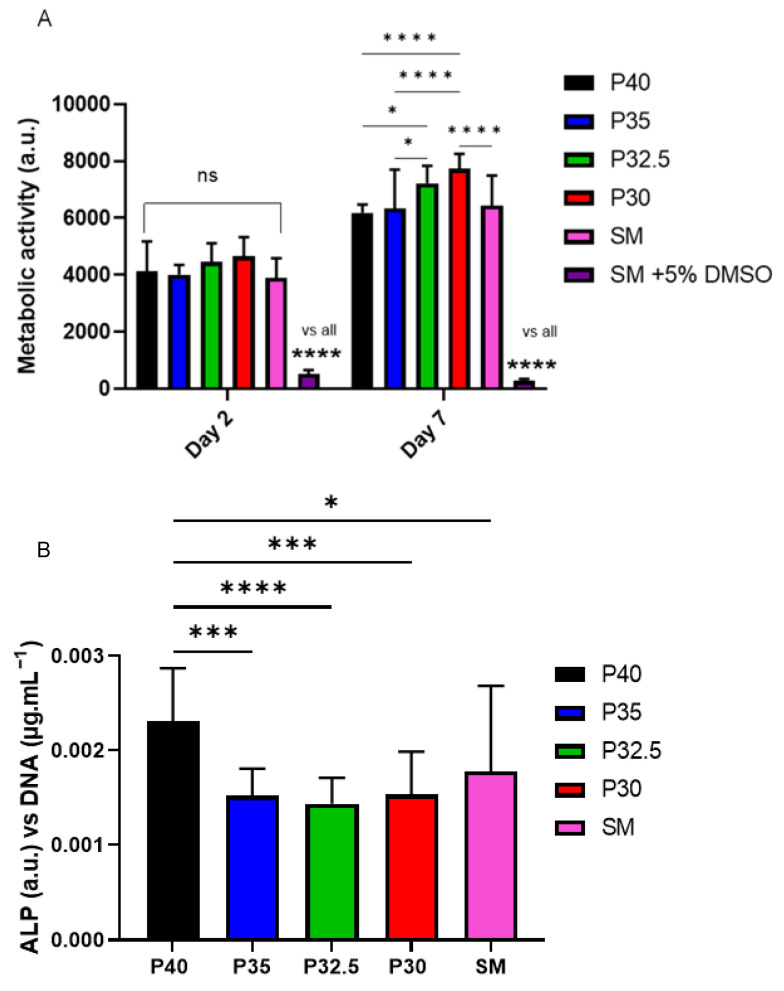
(**A**) Evaluation of cell metabolic activity in indirect culture of porous microspheres at day 2 and day 7. * *p* < 0.05 and **** *p* < 0.0001. (**B**) Evaluation of ALP activity in indirect culture with porous microspheres at day 7. * *p* < 0.05, *** *p* < 0.001 and **** *p* < 0.0001.

**Figure 9 bioengineering-09-00611-f009:**
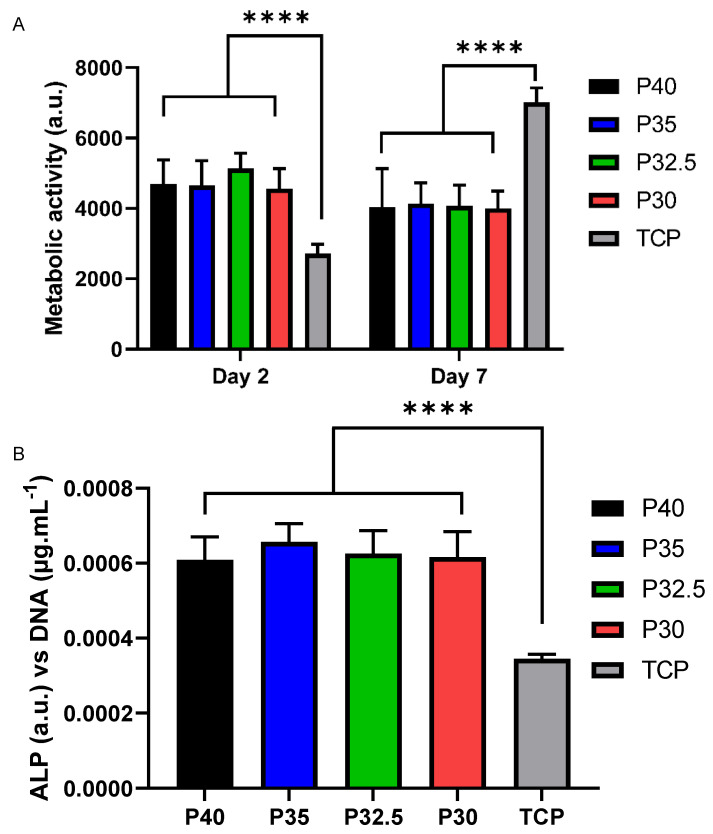
(**A**) Evaluation of cell metabolic activity in direct culture of porous microspheres at day 2 and day 7. **** *p* < 0.0001. (**B**) Evaluation of ALP activity in direct culture with porous microspheres at day 7. **** *p* < 0.0001.

**Figure 10 bioengineering-09-00611-f010:**
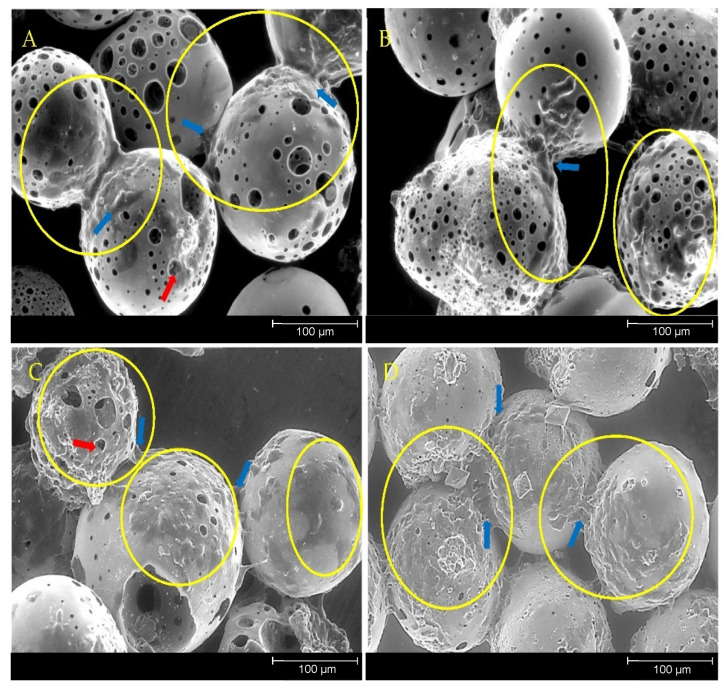
ESEM images of (**A**) P40, (**B**) P35, (**C**) P32.5 and (**D**) P30 PMS after 7 days of direct culture with MG63 cells. Yellow circles indicate regions of cells attachment and the formation of colonies on the microspheres’ surface. Blue arrows indicate cell projections bridging between adjacent microspheres and red arrows to indicate where the cells were penetrating into the pore regions.

**Table 1 bioengineering-09-00611-t001:** Comparison and glass codes used for the four different phosphate glass formulations selected for microsphere preparation and characterisation.

Glass	P_2_O_5_ (mol%)	CaO (mol%)	MgO (mol%)	Na_2_O (mol%)
**P40**	40	16	24	20
**P35**	35	21	24	20
**P32.5**	32.5	23.5	24	20
**P30**	30	26	24	20

## Data Availability

Not applicable.

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
