# Peer review of "Developing Porous Ortho- and Pyrophosphate-Containing Glass Microspheres; Structural and Cytocompatibility Characterisation"

_bioengineering, 2022, doi:10.3390/bioengineering9110611_

Round 1

Reviewer 1 Report

L656-L657 – The authors state that P30 and P32.5 exhibit greater bioactivity and cytocompatibility. This cannot be concluded from the assessment carried out as cell viability was not accessed on metabolic activity.

P801 -803: The authors conclude that the P30 PMS is most favourable. This is not supported by the data in the paper. The P40 shows the highest ALP levels in the indirect culture study. I would suggest that a more indepth cell culture study is required to be able to draw any true conclusions in relation to the optimal group.

L778-780: The authors state that the microspheres porosity facilitates the loading of microspheres with therapeutic cargo for orthobiological applications. There is no evidence of this from the data presented. The authors should rephrase this statement accordingly.

Minor Errors

L126 – ‘that had been kept in at’ replace with ‘that had been kept at’

Figure 3: I presume that the star indicates calcium carbonate? Please indicate in the figure legend.

Figure 6(F): Please amend legend for 32.5

Figure 7 (A): The symbol “ appears on the figure above the XRD pattern for P30. Please remove.

Author Response

We would first like to thank the reviewers for evaluating this manuscript and for the helpful suggestions provided to help improve the manuscript. Please see our responses to reviewers’ comments below (the reviewer’s comments are highlighted in italics).

L656-L657 – The authors state that P30 and P32.5 exhibit greater bioactivity and cytocompatibility. This cannot be concluded from the assessment carried out as cell viability was not accessed on metabolic activity.

Based on the suggestion above, we have modified the statement in the manuscript as follows (which has been highlighted in yellow in the revised manuscript in L661-665

“However, the P30 and P32.5 glass formulations developed which contained high proportions of orthophosphates and pyrophosphates exhibited greater bioactivity and cytocompatibility, which would could lead to more favourable environments to promote bone repair and regeneration than other glasses containing higher phosphate species.”

P801 -803: The authors conclude that the P30 PMS is most favourable. This is not supported by the data in the paper. The P40 shows the highest ALP levels in the indirect culture study. I would suggest that a more indepth cell culture study is required to be able to draw any true conclusions in relation to the optimal group.

As suggested by the reviewer, the following details have been added as amendedment to the above statement (L815-822):

Although no statistically significant differences between the formulations were observed from direct cell culture studies, cells treated with P30-conditioned media revealed higher metabolic activity from the indirect studies. Whilst those treated with P40-conditioned media demonstrated elevated ALP activity. This suggests that a combination of microsphere formulations could be developed to elicit the desired responses. However, further in vitro and in vivo studies would be required to establish whether a specific microsphere formulation or a combination of different formulations would deliver the desired properties for facilitating enhanced bone regeneration.

L778-780: The authors state that the microspheres porosity facilitates the loading of microspheres with therapeutic cargo for orthobiological applications. There is no evidence of this from the data presented. The authors should rephrase this statement accordingly.

 Based on the above comments by the reviewer, the following details were added to the statement (L785) and are highlighted in in the revised manuscript.

Microsphere porosity is desirable as it increases surface area which can allow for greater cell attachment, increased degradation, release of therapeutic ions and can be exploited for encapsulation and release of therapeutic cargo for orthobiologic applications. Previous in vivo studies have shown incorporation of autologous bone marrow concentrate within P40 porous microspheres and other biologics, such as drugs, proteins and growth factors, could also be loaded based on the desired end applications.

Minor Errors

L126 – ‘that had been kept in at’ replace with ‘that had been kept at’

The changes to the above sentence have been made as requested by the reviewer (and highlighted in in the revised manuscript).

Figure 3: I presume that the star indicates calcium carbonate? Please indicate in the figure legend.

The star symbol has been added to the figure legend.

Figure 6(F): Please amend legend for 32.5

The figure has been edited so that all of the P32.5 figure legend is now within the boundary of the figure as requested by the reviewer.

Figure 7 (A): The symbol “ appears on the figure above the XRD pattern for P30. Please remove.

The symbol within the figure has been removed

Reviewer 2 Report

Manuscript ID: bioengineering-1931888

Title: Developing Porous Ortho and Pyrophosphate Containing Glass Microspheres; Structural and Cytocompatibility Characterisation

Authors: Ben Milborne, Lauren Murrell, Ian Cardillo-Zallo, Jeremy Titman, Louise Briggs, Colin Scotchford, Alexander Thompson, Rob Layfield and Ifty Ahmed

This manuscript reports the development of phosphate-based glasses (PBGs) in the series xP2O5·(56-x)CaO·24MgO·20Na2O (mol%), where x=40, 35, 32.5, 30, and their processing into solid microspheres (SMS) and porous microspheres (PMS). The prepared PBGs glasses and their corresponding SMS and PMS were characterized by a range of techniques including XRD, SEM, EDX, mercury porosimetry and 31P MAS-NMR to probe the state of glasses, their chemical composition and the phosphate network connectivity (in terms of the Qn phosphate species), and their evolution upon processing to SMS and PMS. Bioactivity studies were also performed to access the suitability of the PBGs and their SMS and PMS for in vivo orthobiological applications.

The experimental work is of high quality, the data analysis was property done, and the presentation of results and discussion are very clear. This investigation has demonstrated that highly porous phosphate glass microspheres based on the ortho- and pyro-phosphate species are promising materials for bone repair and regeneration. It is recommended that the manuscript be accepted for publication after the authors have considered the following comments.

1. Define “SBF” in line 27 of the Abstract.

2. Lines 41-42: It is stated “The PO43- tetrahedral groups are the structural unit of PBGs, where the phosphorous atom is covalently bonded to three oxygen’s via single bonds and to a terminal oxygen via a double bond”. It is noted that (PO4)3- has 4 non-bridging oxygen’s (NBO) and corresponds to the orthophosphate composition (Q0 species). However, the description given for the phosphorous bonging is appropriate only for the P2O5 glass where the phosphate tetrahedral unit is O=P-(O1/2)3, and this corresponds to neutral species with stoichiometry (PO2.5)0 (Q3 species). Therefore, the above statement should be corrected. It is also suggested to replace the general phrase “PO43- tetrahedral groups” used in the text by the phrase “phosphate tetrahedral groups”, because “PO43- tetrahedral groups” refers only to Q0 species.

3. Lines 52-53: the statement “The addition of alkali (MeO) and alkaline earth (Me2O) oxides” should be corrected to read “The addition of alkali (Me2O) and alkaline earth (MeO) oxides”.

4. The first paragraph of page 6 about the Bruker D8 Advanced X-ray diffractometer (lines 217-223) is the same with the paragraph in the top of page 5 (lines 172-178). One of these paragraphs should be deleted.

5. Page 10, line 326: correct “As seen in Figure 2C” to read “As seen in Figure 2B”.

6. Caption of Figure 3: give symbols for all “crystalline peaks” observed. For example, what is the crystal with peaks marked by stars? 

Author Response

We would like to thank this reviewer for evaluating this manuscript and for the helpful suggestions provided to help improve the manuscript. Please see our responses to reviewers’ comments below (the reviewer’s comments are highlighted in italics).

Define “SBF” in line 27 of the Abstract.                       

SBF has been defined as simulated body fluid in the Abstract.

  1. Lines 41-42: It is stated “The PO43-tetrahedral groups are the structural unit of PBGs, where the phosphorous atom is covalently bonded to three oxygen’s via single bonds and to a terminal oxygen via a double bond”. It is noted that (PO4)3- has 4 non-bridging oxygen’s (NBO) and corresponds to the orthophosphate composition (Q0species). However, the description given for the phosphorous bonging is appropriate only for the P2O5 glass where the phosphate tetrahedral unit is O=P-(O1/2)3, and this corresponds to neutral species with stoichiometry (PO2.5)(Q3 species). Therefore, the above statement should be corrected. It is also suggested to replace the general phrase “PO43- tetrahedral groups” used in the text by the phrase “phosphate tetrahedral groups”, because “PO43- tetrahedral groups” refers only to Q0 species.

We thank the reviewer for this useful insight. As suggested, the following details have been added to the statement and are highlighted in yellow in the revised manuscript.

The phosphate tetrahedral groups are the structural unit of PBGs, where the phosphorous atom is covalently bonded to three oxygen’s via single bonds and to a terminal oxygen via a double bond. with P-O-P bonds that form between adjacent tetrahedral are known as bridging oxygens (BOs) [3]. The structure of PBGs is related to the number of BO’s and Qn terminology is used to describe the species present within PBGs, where n signifies the number of BO’s per PO43- tetrahedron [4]. A range of phosphate glass anionic Qn species can be produced from Q3, Q2, Q1 and Q0 species which are referred to as ultra-, meta-, pyro- and orthophosphates respectively.

  1. Lines 52-53: the statement “The addition of alkali (MeO) and alkaline earth (Me2O) oxides” should be corrected to read “The addition of alkali (Me2O) and alkaline earth (MeO) oxides”.

We thank the reviewer for spotting this error. As suggested, the following details have been corrected and added to the statement as highlighted in yellow below.

“The addition of alkali (Me2O) and alkaline earth (MeO) oxides”.

  1. The first paragraph of page 6 about the Bruker D8 Advanced X-ray diffractometer (lines 217-223) is the same with the paragraph in the top of page 5 (lines 172-178). One of these paragraphs should be deleted.

The paragraph on page 6 has now been deleted. The different parameters used for XRD analysis of porous microspheres immersed in SBF have been added to page 5 so that it now reads as follows:

“A Bruker D8 Advanced X-ray diffractometer (Bruker-AXS, Karlsruhe, Germany) was used to determine the amorphous nature of the ground phosphate glass samples at room temperature using a Ni-filtered Cu-Kα radiation source. Data points were obtained every 0.02° from 10-70° over a 10 minute period. The resulting data was analysed using DIFFRAC.EVA software (DIFFRAC-plus suite, Bruker-AXS) to identify phases through a database of known peaks from the International Centre for Diffraction Data (ICDD) database 2005. When studying porous microspheres following SBF immersion, data points were obtained every 0.01° from 5-70° over a 4 hour period with a step size of 2.2 seconds.

  1. Page 10, line 326: correct “As seen in Figure 2C” to read “As seen in Figure 2B”.

As suggested by the reviewer, the sentence now reads “As seen in Figure 2B”.

  1. Caption of Figure 3: give symbols for all “crystalline peaks” observed. For example, what is the crystal with peaks marked by stars? 

The star symbol has now been added to the figure legend to indicate calcium carbonate which was used as the porogen material.

Reviewer 3 Report

In this manuscript entitled “Developing porous ortho and pyrophosphate containing glass microspheres; structural and cytocompatibility characterization”, the authors show the usefulness of the porous phosphate-based glasses for bone repair and regeneration materials. Although they lack in vivo experiment, the data presented are strong and convincing evidence. I think this is an interesting manuscript. 

The topic is very important and interesting but all sections, except for “2. Materials and Methods”, in the text page of this article is too redundant. The authors should try for the shortening of the article.

Author Response

We would also like to thank this reviewer for evaluating this manuscript and for the suggestions provided to help improve the manuscript. Please see our responses to reviewers’ comments below (the reviewer’s comments are highlighted in italics).

In this manuscript entitled “Developing porous ortho and pyrophosphate containing glass microspheres; structural and cytocompatibility characterization”, the authors show the usefulness of the porous phosphate-based glasses for bone repair and regeneration materials. Although they lack in vivo experiment, the data presented are strong and convincing evidence. I think this is an interesting manuscript. 

The topic is very important and interesting but all sections, except for “2. Materials and Methods”, in the text page of this article is too redundant. The authors should try for the shortening of the article.

The authors have reviewed the manuscript again and in accordance to the comments raised by the other reviewers. We have attempted to reduce redundant text wherever possible.

Reviewer 4 Report

This study addresses an interesting issue, is methodologically sound and has an extensive experimental work, being well written and easy to follow. However, considering the extensive knowledge in this issue, it is not clear to the reader what is the real added value / novelty of this study. It would be helpeful if the authors  can emphasize this.

Author Response

We are also grateful to this reviewer for evaluating this. Please see our responses to reviewers’ comments below (the reviewer’s comments are highlighted in italics).

This study addresses an interesting issue, is methodologically sound and has an extensive experimental work, being well written and easy to follow. However, considering the extensive knowledge in this issue, it is not clear to the reader what is the real added value / novelty of this study. It would be helpful if the authors can emphasize this.

We thank the reviewer for raising this point. The following details have been added to the revised manuscript as highlighted in yellow (L784-790) below. This addition emphasises the novelty of the study which was mainly due to the production (for the first time) of porous PBG glass microspheres from ortho and pyrophosphate rich formulations.

This work demonstrated that highly porous phosphate glass microspheres containing only ortho and pyrophosphate species are highly promising materials for bone repair and regeneration. This study shows for the first time that these novel ortho and pyrophosphate rich glass formulations could be processed into solid and porous microspheres. Microsphere porosity is desirable as it increases surface area which can allow for greater cell attachment, increased degradation, release of therapeutic ions and can be exploited for encapsulation and release of therapeutic cargo for orthobiologic applications. Previous in vivo studies have shown incorporation of autologous bone marrow cells within P40 porous microspheres and other biologics, such as drugs, proteins and growth factors, could also be loaded within the microspheres. The simple and rapid manufacturing process developed can produce high yields of porous microspheres with fully interconnected porosity using the flame spheroidisation method. The porous microsphere formulations investigated in this study warrant further investigation to evaluate their effect on osteoconduction and osteoinduction and the formation and maturation of new bone tissue in vivo.

Round 2

Reviewer 3 Report

The manuscript has been modified appropriately. I can admit to accept this article.